# The Microemulsion with Solubilization of the Ethanolic Extract of the Leaves of Melão-de-São-Caetano (*Momordica charantia*) and Antibacterial Action

**DOI:** 10.3390/jfb14070359

**Published:** 2023-07-10

**Authors:** Aline M. Q. de Brito, Wilka da Silva Camboim, Cátia Guaraciara F. T. Rossi, Ivan A. de Souza, Késia K. O. S. Silva

**Affiliations:** 1Technology Center, Postgraduate Program in Textile Engineering (PpgET), Federal University of Rio Grande do Norte, Natal 1524, RN, Brazil; aline_meiryelle@hotmail.com (A.M.Q.d.B.); wilkacamboim@gmail.com (W.d.S.C.); kesia.souto@ufrn.br (K.K.O.S.S.); 2Natalense University Center (UNICEUNA), Natal 4890, RN, Brazil; catia_gua@yahoo.com.br; 3Technology Center, Plasma Materials Processing Laboratory (LABPLASMA), Federal University of Rio Grande do Norte, Natal 1524, RN, Brazil

**Keywords:** *Momordica charantia*, microemulsion, functionalization, antibacterial

## Abstract

Extracts obtained from plants have significantly contributed to the creation of new drugs due to their medicinal properties, which are provided by the presence of bioactive components. This has led to a growing interest from the pharmaceutical industry in using this type of extract for the creation of increasingly advanced medications. The main components sought are antibacterial agents from sustainable and renewable sources, whether of animal or vegetable origin or derived from other natural components. Tissues become a source of microbial proliferation, especially when in contact with the human body, which can cause serious diseases. In line with this, the goal of this research was to create an antibacterial Melon-de-São-Caetano (*Momordica charantia*) leaf microemulsion for application on material surfaces. This microemulsified system is an effective alternative for solubilizing functional agents, and being thermodynamically stable, it is efficient for long-term use. For this study, an extract of *Momordica charantia* leaves (EMC) was obtained, and microemulsions with different EMC concentrations (P1, P2, and P3) were produced. The extract and microemulsions were investigated using Fourier Transform Infrared (FTIR) spectroscopy, particle size, zeta potential, thermal stress, pH, electrical conductivity, Transmission Electron Microscopy (TEM), and antibacterial analysis (*Staphylococcus aureus*). In summary, the proposed objective was met, and EMC, SME, and the P2 and P3 microemulsions showed positive results against S. aureus, with the P3 microemulsified system being the most effective with a 12.5 mm inhibition halo. Therefore, the product developed in this research has the potential for application on surfaces, providing antibacterial action.

## 1. Introduction

Clothing provides a protective barrier for the body against environmental variations, such as wind, heat, moisture, etc. These variations create an excellent environment for the proliferation of pathogenic microorganisms such as *Staphylococcus aureus* bacteria on the body. Although these bacteria are part of the skin’s microbiota, excessive growth can trigger diseases, especially in people who are constantly exposed to sweat, as is the case with athletes in general [1].

Products with antibacterial effects have become a necessity in the field of research and application in both the medical and sports fields, as well as in other sectors, due to the possibility of preventing the proliferation of bacteria such as Staphylococcus aureus. Concurrently, there is an urgent need for this product to have sustainable and environmentally friendly characteristics, such as using plant-based derivatives (plant extracts) [2,3,4].

A commonly used plant for obtaining plant extracts is *Momordica charantia* (MC), which is found in broad biodiversity and is popularly known as bitter melon or bitter gourd and belongs to the Cucurbitaceae family, and it has become popular since ancient times due to the discovery of its application in medicine [5].

The biological properties of the bitter melon plant are attributed to the existence of substances such as tannins, flavonoids, saponins, cardiac glycosides, and steroids, which are natural bioactive compounds [6]. The biological properties of the bitter gourd plant are due to the presence of substances such as tannins, flavonoids, saponins, cardiac glycosides, and steroids, which are natural bioactive agents. In addition, phytochemical analyses also show the presence of various components that have active agents in different parts of the *Momordica charantia* plant. The leaves are considered a rich source of biological constituents, as they contain phenolic compounds that include caféic acid, gallic acid, rutin, catechin, genotoxic acid, o-coumaric acid, ferulic acid, luteolin, apigenin, and thyme [7,8,9].

Antibacterial activity was evaluated in the aqueous and ethanolic extracts of the leaves of this plant, and both showed activity against the bacteria *Staphylococcus aureus*, *Streptococcus mutans*, *Bacillus subtilis*, *Escherichia coli*, and *Pseudomonas aeruginosa* [10,11]. Another recent study investigated the antibacterial properties of Staphylococcus aureus with the extracts from different parts of the plant with the solvents hexane and ethanol, where the result showed that the leaf extract using hexane presented an ineffective effect against the bacterium, but in the case of the ethanolic extract, the inhibition halo was 34 mm, which confirms the sensitivity of *S. aureus* to the ethanolic extract of MC leaves [12].

Natural extracts can also be part of the microemulsion system, which comprises nanoscale substances used to deliver bioactive agents to the surface of materials. The advantage of using these systems is their ease of preparation and application, as they have the ability to form spontaneously. This makes it an appropriate system for increasing the absorption rate of active agents. Furthermore, it has excellent solubilization capacity for lipophilic drugs and is thermodynamically stable, making it suitable for long-term use [13,14,15]. The extract of *Momordica charantia* leaves, along with Tinospora cordifolia and Trigonell foenum graecum, was solubilized in a pseudo ternary microemulsion system with a mean particle size of 861nm and good physical stability according to thermal stress analysis. The UV spectrum also showed that the sample was nearly transparent by verifying the transmission percentage [12,16].

Therefore, this investigation aims to obtain the ethanol extract of the leaves of *Momordica charantia* and microemulsion systems with different concentrations of this solubilized extract in order to analyze the best composition from the phase diagram of this system and to verify the best concentration of this extract to produce the effect against *Staphylococcus aureus*.

## 2. Materials and Methods

### 2.1. Obtaining the Ethanol Extract

Translation: The *Momordica charantia* leaves used in this study were collected in the city of São Gonçalo do Amarante, Rio Grande do Norte. They were washed in running water and placed in an oven at 35 °C for 16 h to dry. After that, the maceration and sifting process was performed at room temperature. Then, the solution was prepared by adding the particulate leaves together with 70% ethanol at a ratio of 1:5. This mixture was then allowed to rest for 24 h at room temperature, and when filtered, a new sample was obtained. Finally, the solution was subjected to evaporation at an approximate temperature of 60 °C for approximately 3 h and heated at a temperature of 35 °C for 18 h, resulting in the ethanolic extract of *Momordica charantia* leaves (EMC). This methodology was based on the study of Nagarani, Abirami, and Siddhuraju [17]. The EMC was then diluted in an aqueous medium at a concentration of 0.2 g/mL. This solution was constantly stirred on the SL-91 magnetic stirrer model SOLAB at room temperature, thus obtaining the mother solution of the extract (SME).

### 2.2. Obtaining the Phase Diagram

A microemulsion is only obtained if the amounts of the components are carefully quantified in a mixture [18]. Thus, the preparation of a phase diagram is essential for choosing the concentration of each substance. For this study, bidistilled water, soy oil, and the surfactant Tween^®^ 80 were used. Nine points were made using the mass titration method with these three components, where the masses of the surfactant + oil binary were fixed, and water is titrated to it until there is a visual change in the system from clear to turbid or vice versa. The components were weighed using an analytical balance (AUY220—Shimadzu) and then placed in the magnetic stirrer with heating (SL—91, SOLAB model) at a temperature of 60 °C.

### 2.3. Choice of Constituents and Preparation of the Microemulsified System

The choice of the commercial surfactant, Tween 80, was made due to its non-ionic character. These surfactants promote better stability in the microemulsion. This effect is due to the lesser interaction that exists between the surfactant heads in the micelles because they do not have ions. Furthermore, Tween 80 is low-cost and biodegradable, making it one of the most commonly used surfactants in micro/nanoemulsified systems [14,19].

The choice of soybean oil was made for the oily phase because it is an inert vegetable oil. In addition, its presence in formulations promotes an increase in the compatibility of the oily phase with non-ionic surfactants [20]. Finally, the choice of the aqueous phase was distilled water due to being a natural and low-cost solvent.

Four microemulsions were prepared; initially, a base microemulsion (M) was prepared, which did not contain the extract, being composed of Tween 80, soybean oil, and water.

Thus, the elaboration of the other MEs was made from sample M together with the SME with varying concentrations of 0.5%, 1%, and 5%, and was denoted as P1, P2, and P3, respectively. Thus, points 1, 2, and 3 were produced as follows:Point 1 (Concentration 0.005 g/mL): 50 g of base microemulsion and 0.25 g of SME;Point 2 (Concentration 0.01 g/mL): 50g of base microemulsion and 0.5 g of SME;Point 3 (Concentration 0.05 g/mL): 50g of base microemulsion and 2.5 g of SME.

### 2.4. Particle Size and Zeta Potential

The sizes of the suspended particles for this research were determined by dynamic light scattering using the Nanotrac 252 equipment. The zeta potential technique is a tool for understanding surface charges on a nanoscale and predicting their stability. Knowing that to observe the stability of an emulsion, it is necessary to perform a zeta potential analysis (as stability depends on both the polarity of the surfaces and the presence of ionic structures), the determination of the zeta potential was conducted similarly to that performed by Varenne et al. [21], on the Zeta Potential Analyzer equipment, model Zetaplus, from the manufacturer Brookhaven Instruments Corporation, with the ZetaPlus software. These two variables provide information about the physical stability of the produced microemulsions.

### 2.5. Thermal Stress

This technique evaluates the moment at which the formulations become unstable when exposed to temperature variations. To verify the thermal stability of the microemulsions (M, P1, P2, and P3), based on the methodology used by Gupta et al. [22], each sample was placed in a vial with a 5 mL aliquot. The vials were then heated to 65 °C at room temperature using an SL-91 magnetic stirrer (SOLAB model), and temperatures were monitored by a K29 S090 thermometer. This work was performed in triplicate, so the samples were analyzed taking into account the occurrence of phase separation or any visual changes.

### 2.6. Hydrogenionic Potential (pH)

The pH value in a microemulsion system is critical in evaluating their preservation and thus verifying any possible interactions, incompatibilities, and stabilities. The determination of pH for MEs (M, P1, P2, and P3) was performed based on ABNT NBR 7353, using a portable pH meter model, mPa-210P, from MStecnopon equipment LTDA. The measurement was made by inserting the measuring cell directly into the samples at room temperature and was performed in triplicate.

### 2.7. Electric Conductivity

The electrical conductivity measurements in the MEs measure the capacity of the medium to allow the continuous flow of electrical current in oil or in an aqueous medium. Therefore, the analysis of the electrical conductivity of the systems (M, P1, P2, and P3) was checked 24 h after the development of the microemulsions by inserting the electrode according to ABNT NBR 14340, and the readings were taken in room temperature in triplicate with the Portable Digital Conductivity Meter AKSO, model AK87.

### 2.8. Dynamic Viscosity

For this study, the viscosity of the produced MEs, M, P1, P2, and P3, were analyzed on the Anton-Paar Viscodensimeter, model SVM 3000. The tests were performed in triplicate at a temperature of 20 °C according to ASTM D 7042. The equipment was calibrated, and then the samples were inserted into it using a 5 mL syringe.

### 2.9. Fourier Transform Infrared Spectroscopy (FTIR)

The FTIR technique was used to observe the presence of the absorption bands of the functional groups present in the samples. For this research, this was performed for the SME, M, P1, P2, and P3 samples, and for this purpose, a Shimadzu Fourier Transform Infrared Spectrophotometer, model IRAffinity-1, was used in an ATR horizontal PIKE Technologies system, model MIRacle, within the range of 400–4000 cm^−1^.

### 2.10. Transmission Electron Microscopy (TEM)

In this study, high-resolution transmission electron microscopy (HR-TEM) was used to observe the morphology of the microemulsified systems. HR-TEM allows the obtainment of high-resolution images for determining particle size and particle system polydispersity. In this investigation, the technique was performed for sample P3 with the highest concentration of EMC, since the components of samples P1, P2, and P3 were the same, with only different concentrations. The microscope used was a JEOL—2100 model equipped with EDS and Thermo Scientific.

### 2.11. Antibacterial Analysis

The verification of the antibacterial activity of the products developed in this study was carried out in the Multidisciplinary Laboratory of Facisa (Faculty of Health Sciences—UFRN), where samples of the concentrated extract of M. Charantia leaves (EMC), SME, M, P1, P2, and P3 were placed in contact with the inoculated agar plate with the bacterium Staphylococcus aureus (AATCC 25923). Thus, after 24 h of incubation, the plates were observed with the aim of analyzing the action of the microemulsions against bacteria, wherein it is necessary to present an inhibition zone of growth around the sample. Thus, the size of this inhibition zone will be strongly influenced by the diffusion capacity of the antibacterial agent.

## 3. Results and Discussion

### 3.1. Fourier Transform Infrared Spectroscopy

The FTIR showed the presence of the main functional groups and their variation, as shown in Figure 1, where the FTIR curves are represented for the base microemulsion (tween 80 + soybean oil + water)—M, in black; the mother solution of the extract—SME (red curve); microemulsion Point 1—P1 (blue curve); microemulsion Point 2—P2 (green curve); and microemulsion Point 3—P3 (magenta curve).

For the mother solution of the extract from the leaves of the “melon-de-são-Caetano” (SME), the absorption bands shown in Figure 1 shows the absorption bands at 1125 cm^−1^, 1630 cm^−1^, 2104 cm^−1^, 3325 cm^−1^.

So, from Graph 2, the absorption band at 3325 cm^−1^ is observed, which corresponds to the O-H stretching due to the presence of water. The peak at 2104 cm^−1^ occurs due to the presence of -CH3, while the band at 1630 cm^−1^ is related to the stretching of the functional group C = C. The characteristic peak at 1125 cm^−1^, which is shown subtly on the graph, is due to the stretching of C-N. The absorption band that is presented at 615 cm^−1^ is due to the functional group C-H out of the plane. These absorption bands are due to the following components: proteins, triterpenes, alkaloids, steroids, and phenols, which are compounds present in the M. charantia leaf extract, supporting the phytochemical studies present in the literature [23,24,25].

Furthermore, according to the authors Krithika and Bridget [26], The absorption bands presented at 2104 cm^−1^ are due to the presence of the compound mormodicine, which is a type of terpenoid found in the leaves of MC, and the absorption band presented at 1630 cm^−1^ corresponds to the component Charantina, which is a triterpene also found in the leaves of this plant.

The functional groups present in the base microemulsion (M) are shown on the graph, which displays the absorption bands evidenced for this sample, being 3335 cm^−1^, 2125 cm^−1^, 1645 cm^−1^, and 1080 cm^−1^, respectively.

According to the results for formulation M, four absorption bands are evident. In this case, the large absorption band is observed at the peak at 3335 cm^−1^, which corresponds to the presence of water in the microemulsions and is characterized by the stretching of O-H. The band evidenced at 2125 cm^−1^ is due to the presence of the functional group -CH3, one of those responsible for the alteration of the pH of the solutions. In the case of the peak at 1645 cm^−1^, it represents the stretching of the functional groups C=O, which is characteristic of the ester compound. The absorption band present at 1080 cm^−1^ confirms the stretching of the C-O-C chain, which corresponds to the carbon ether constituent and the latter, present in the non-ionic surfactants [27]. The microemulsions P1, P2, and P3 evidenced the same functional groups present in sample M with varying intensity; this is due to the strong compatibility between the SME and the base microemulsion [16]. Table 1 shows the comparison of the functional groups of the SME, M, P1, P2, and P3 samples.

### 3.2. Phase Diagram and Microemulsions

According to the data observed in Figure 2, this microemulsified system showed two Winsor regions and a WII region, where the water phase is in excess, forming a two-phase system and a WIV region characterized by a single phase (the microemulsion), thus confirming that this system is Oil/Water O/W [28].

Therefore, for this study, we chose a point in the WIV region, highlighted in red, which represents the concentrations for the formulation of sample M, corresponding to 20% surfactant, 1% oil phase, and 79% water phase.

The results of this diagram made it possible to find a suitable region in which the composition of the elements is most favorable in terms of viscosity and composition, with a view into the application of these systems.

Thus, after the microemulsion formation process and according to the methodology determined by the phase diagram, 25 mL of the four microemulsions were obtained, and the base microemulsion (M) and the microemulsions with SME solubilized at concentrations P1, P2, and P3, with 0.5%, 1%, and 5%, respectively, as shown in Figure 3.

### 3.3. Particle Size, Zeta Potential, Thermal Stress, and Dynamic Viscosity

Thus, the properties of the materials are influenced by the particle size range, the zeta potential values, and the temperature resistance, making it important to understand these behaviors for possible applications. The four microemulsions (M, P1, P2, and P3) underwent these characterization analyses, and their values can be seen in Table 2. According to the particle size values displayed in Table 2, it is observed that the base microemulsion has the smallest particle size of 8.86 nm, and for the microemulsions with the solubilized SME, it is possible to verify that the increase in the concentration of this constituent also increases the droplet size. However, all the MEs have values lower than 20 nm, which makes these results favorable for application, as values below this particle size value improve the delivery of the formulation in the substrate since it allows the light to scatter less than the wavelength of light; thus, the microemulsion remains clear and optically transparent [29].

The evaluation of the zeta potential for the four microemulsions (M, P1, P2, and P3) ranged between −6.80 and −9.00 mV. We can relate this change in the potential to the particle size by observing that the progressive increase in particle size did not have the same effect on its charge; this probably happened due to the decrease in pH (see Table 3), where we know that this occurs due to the change in the amount of H+ ions in the solution. Additionally, for the particles of a formulation for microemulsions to be considered stable [30], the zeta potential data must comprise values that are between +30 mV and −30 mV. Therefore, the values found for M, P1, P2, and P3 support the data from the previously cited literature, and, therefore, the physical stability of the four microemulsions under study is justified.

The formulated microemulsions had the phase separation temperature limit from the thermal stress that started at room temperature, and the phase change for each of the samples occurred 1 °C above the temperatures shown in Table 2. In this way, samples M, P1, P2, and P3 showed resistance and remained stable up to temperatures of 61.3 °C, 48.6 °C, 43.3 °C, and 41.9 °C, respectively. It is also observed that as SME is added and its concentration is increased, there is an instability of the formulations [31] that leads to a decrease in resistance to thermal stress; this is because when SME is added to M, we are altering the number of solutes because, as we saw, the particle sizes increase with the contraction of such a solution, which in turn alters the viscosity that is also altered by the presence of a larger amount of water in the solution, which in this case caused a lower temperature resistance for the less dense solution.

### 3.4. Hydrogenionic Potential (pH), Electrical Conductivity

We can also relate the results of the pH, electrical conductivity, and dynamic viscosity measurements to their respective values listed in Table 3.

In systems that use natural extracts, the pH value of the formulation is usually decreasing, making it more acidic. According to Table 3, it is observed that the presence of SME in microemulsion systems and the increase in its concentration decreases the pH value; this occurs due to the presence of phenolic groups, which are characteristic of O-H chemical bonds. This way, hydrogen ionization occurs in the hydrogen present in the phenolic group with the hydrogen in the aqueous medium, making it a medium with a higher index of acidity due to the increase in the amount of H+ [32]. All of this is in line with the variations of the zeta potential seen earlier.

Electrical conductivity was investigated in order to confirm the presence of charged ions in the microemulsions. According to the data presented in Table 3, it is evident that with the insertion of the SME and the increase in its concentration, the electrical conductivity values were increased, which can be explained by the presence of H+ ions in the water that increased with the addition of PME. Thus, the four microemulsion systems have high electrical conductivity due to the high concentration of the aqueous phase rich in ions, whether from solvents or solutes in the microemulsion systems [33,34], and the behavior of the previously described phenomena can be easily understood by analyzing Figure 4A,B.

### 3.5. Transmission Electron Microscopy

The high-resolution transmission electronic microscope produced images obtained from the P3 sample, i.e., which contains 5% of the SME. Thus, through the average size of the particles and the micrograph with a magnification of 5 μm in Figure 5, one can observe several particles agglomerated in nanometric sizes, with an average of 11.28 nm ± 1.09 nm; also, these particles have a spherical shape [35].

### 3.6. Antibacterial Analysis

The EMC, SME, M, P1, P2, and P3 samples were tested in order to evaluate their effects against the bacterium *Staphylococcus aureus* (Gram-positive bacteria), which, according to the national health surveillance agency (ANVISA), is generally responsible for infections in the skin. In this context, the samples were observed after 24 h of incubation and showed their *S. aureus* growth inhibition halos, as shown in Figure 6, and the average sizes of the formed halos, in mm, in Table 4.

This analysis evaluated the activity against S. aureus of the samples developed in this study, which showed an efficient result for EMC and SME, as shown in Figure 6A, and also for microemulsions P2 and P3, as shown in Figure 6B. Thus, the inhibition halo presented by the crude extract of MC leaves corroborated the results obtained by [7,12,36]. This has a possible explanation from the analysis of the functional groups present in Figure 1. Furthermore, it was seen that in EMC and SME, the change in the number of ions led to an increase in the pH of these solutions, which is one of the factors responsible for the bacteria, as bacteria in acidic or basic media tend not to proliferate; in our result, this is confirmed because samples P2 and P3 had a pH value below 6, making the media slightly acidic (see Table 4) [37].

The M and P1 microemulsion systems did not show inhibition, and this occurred because the base microemulsion is inert and does not have a bioactive action; for the P1 sample, the low concentration of SME was not able to inhibit *S. aureus* because a pH value above 6 (almost neutral) would not have inhibitory properties. However, samples P2 and P3 showed efficient activity, with P3 having the highest inhibition zone, as it is the microemulsion with the highest concentration (5%) of SME.

## 4. Conclusions

Obtaining the extract and mother solution of Melon-de-São-Caetano leaf extract was efficient and optimized, wherein micellar formation took place; however, the Charantin compound identified in the FTIR analysis has saponins with a short non-polar part, which makes it difficult to interact with vegetable oil, which has a long non-polar part. Therefore, it is necessary to solubilize this SME in a microemulsion system. Through particle size, zeta potential, electrical conductivity, and TEM techniques, it was verified that samples M, P1, P2, and P3 were stable systems, and thus the particle sizes are suitable for use in formulations for the vehicle of active agents. According to the exposure to different temperatures, samples M, P1, P2, and P3 showed thermal stress at temperatures of 61.3 °C, 48.6 °C, and 41.9 °C, respectively. Thus, it is necessary to apply these formulations through a process at low temperatures, allowing low energy expenditure. The antibacterial analysis shows the potential of the EMC. Antibacterial action was also verified for microemulsions P2 and P3. In this way, this research presents an antibacterial product with a sustainable character, being a potential alternative for its application on surfaces.

## Figures and Tables

**Figure 1 jfb-14-00359-f001:**
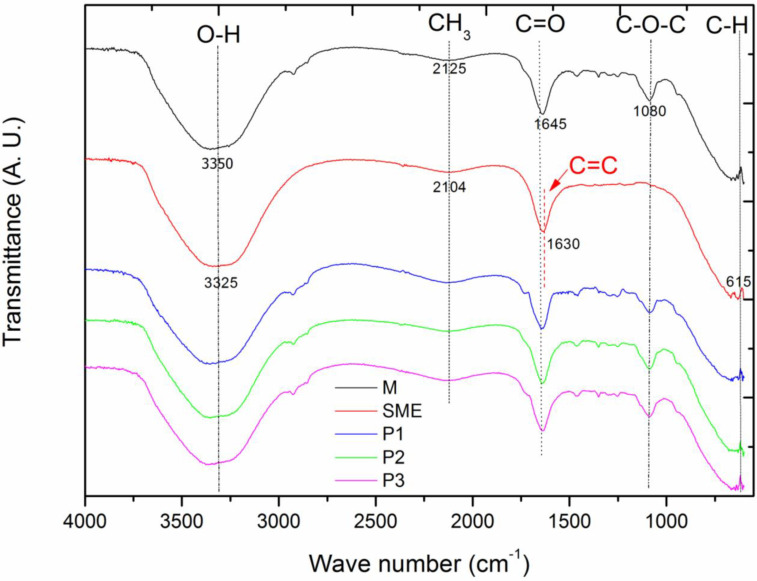
FTIR analyses for the base microemulsion—M (black curve), the mother solution of the extract—SME (red curve), microemulsion Point 1—P1 (blue curve), microemulsion Point 2—P2 (green curve), and microemulsion Point 3—P3 (magenta curve).

**Figure 2 jfb-14-00359-f002:**
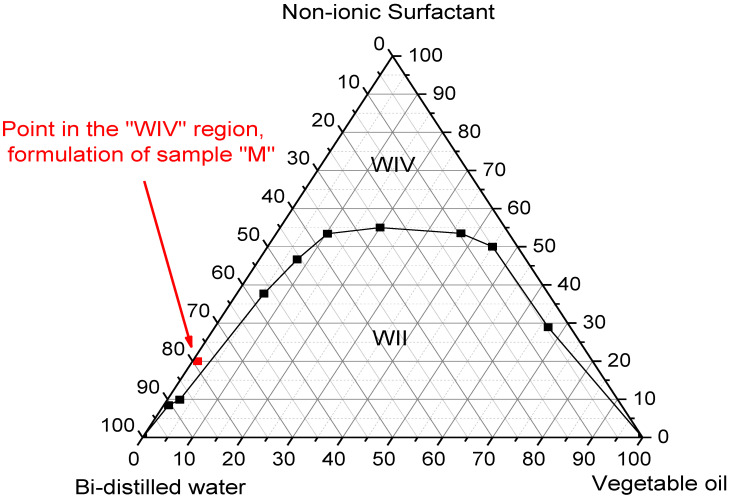
Ternary phase diagram.

**Figure 3 jfb-14-00359-f003:**
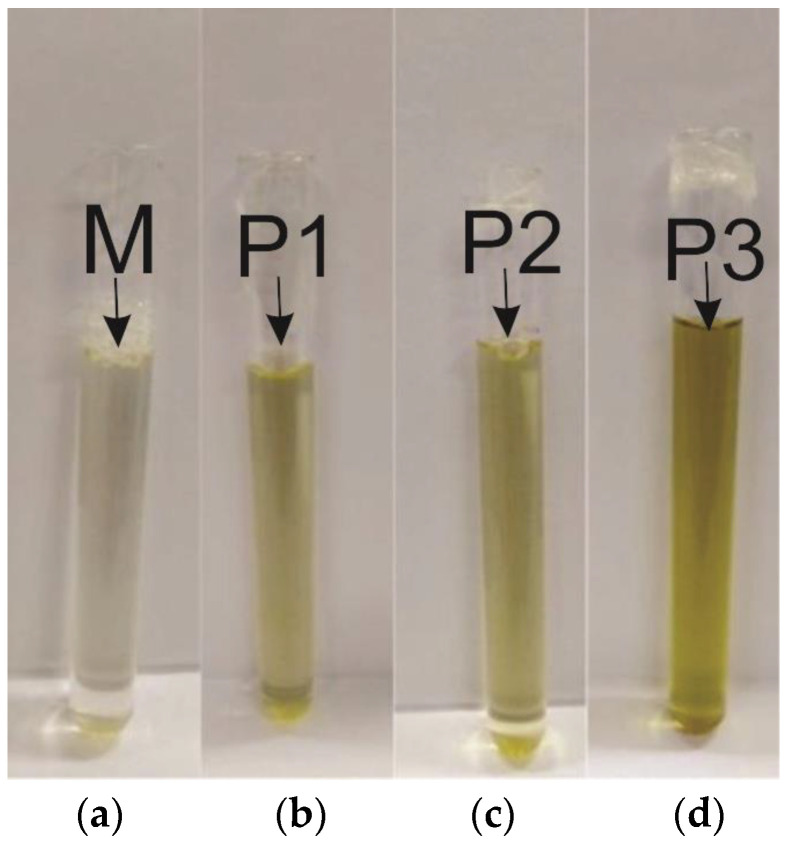
Developed microemulsions: (**a**) Base microemulsion; (**b**) Point 1; (**c**) Point 2; (**d**) Point 3.

**Figure 4 jfb-14-00359-f004:**
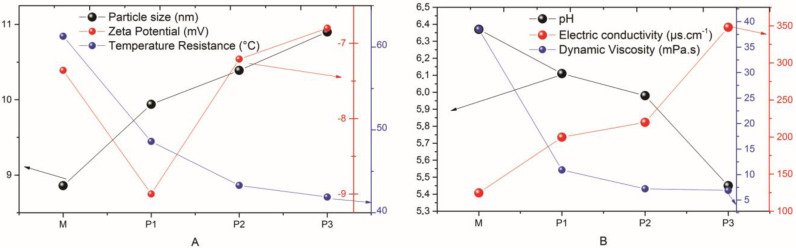
The behavior of the studied parameters. (**A**) Particle size, zeta potential, and temperature resistance; (**B**) pH, electric conductivity, and dynamic viscosity.

**Figure 5 jfb-14-00359-f005:**
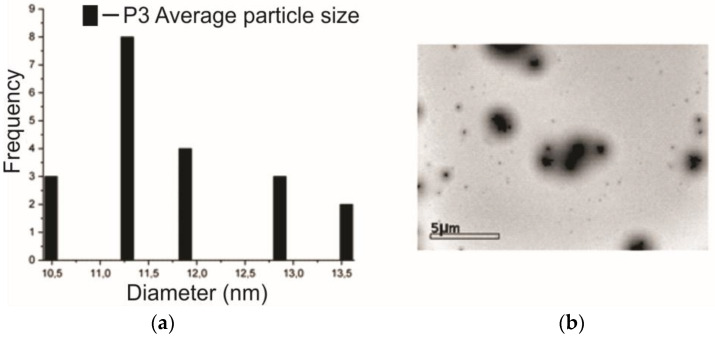
(**a**) Details of the observed particle size and (**b**) micrograph of the P3 sample.

**Figure 6 jfb-14-00359-f006:**
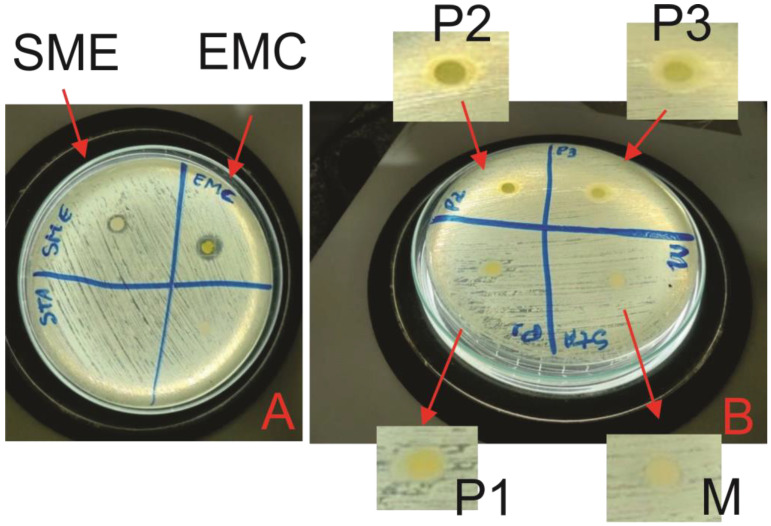
Antibacterial activity: (**A**) EMC and SME; (**B**) M, P1, P2, and P3.

**Table 1 jfb-14-00359-t001:** Main functional groups of the SME, M, P1, P2, and P3 samples.

Samples	Main Groups Functional	Wave Number (cm^−1^)
SME	O-H	3325
	CH_3_	2104
	C = C	1630
	C-N	1125
	C-H	615
M	O-H	3335
	CH_3_	2125
	C = O	1645
	C-O-C	1080
P1 P2 P3	O-H	3325–3335
	CH_3_	2110–2125
	C = O	1635–1645
	C-O-C	1080–1095

**Table 2 jfb-14-00359-t002:** Values of particle sizes, zeta potential, and temperature for microemulsions M, P1, P2, and P3.

Samples	Particle Size (nm)	Zeta Potential (mV)	Temperature Resistance (°C)
M	8.86 ± 1.21	−7.36 ± 0.91	61.30 ± 0.99
P1	9.94 ± 1.59	−9.00 ± 0.43	48.60 ± 1.74
P2	10.39 ± 1.83	−7.21 ± 0.87	43.30 ± 1.56
P3	10.90 ± 1.44	−6.80 ± 0.95	41.90 ± 1.65

**Table 3 jfb-14-00359-t003:** The pH, electrical conductivity, and dynamic viscosity of microemulsified systems M, P1, P2, and P3.

Samples	pH	Electric Conductivity (µs.cm^−1^)	Dynamic Viscosity (mPa.s)
M	6.37 ± 0.34	124.7 ± 0.70	38.50 ± 1.51
P1	6.11 ± 0.15	199.7 ± 0.58	10.90 ± 0.02
P2	5.98 ± 0.28	219.7 ± 0.94	7.20 ± 0.05
P3	5.45 ± 0.42	348.0 ± 047	6.90 ± 0.06

**Table 4 jfb-14-00359-t004:** Inhibition halo against *S. aureus* inhibition halo for EMC, SME, M, P1, P2 e P3.

Samples	Halo of Inhibition against *S. aureus* (mm)
EMC	13.0
SME	9.5
M	0.0
P1	0.0
P2	9.5
P3	12.5

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
