# Peer review of "The Microemulsion with Solubilization of the Ethanolic Extract of the Leaves of Melão-de-São-Caetano (Momordica charantia) and Antibacterial Action"

_jfb, 2023, doi:10.3390/jfb14070359_

Round 1

Reviewer 1 Report

In this manuscript, extract of Momordica charantia leaves and its microemulsions were obtained and the characteristics were studied including FTIR, spectroscopy, particle size, zeta potential, thermal stress, pH, electrical Conductivity, TEM, and antibacterial analysis. The experimental design and result analysis are logical. The overall work is robust and interesting. However, here are some suggestions below to improve the manuscript:

1. Line 35-40, what is the connection between the first paragraph of the introduction and the title article and the following part? No logical relationship seems to be expressed.

2. Line 70-80, paragraphs 6-7 of the introduction can be merged into one paragraph, and a few more examples of Momordica charantia leaves extract used in microemulsion system can be given to show the significance of this study.

3. Line 61-63, the authors state that Momordica charantia leaves extract has antibacterial effect on some bacteria, but why only Staphylococcus aureus was selected in this manuscript? Why didn't it do an antibacterial effect on other bacteria?

4. Line 196 and Line 318, please revise the title of section 3.1 and 3.5, currently this statement is not suitable for this part of the topic.

5. Line 197-201, what results are indicated by the comparison of each curve in Figure 1?

6. Line 203 and 209, What does "graph 2" mean?

7. Line 244, why choose this key point and is there any basis for the choice of the location of this point?

8. Line 338-339, can the inhibition results in this section be compared to those in other studies, such as those cited here 7,12,36 or other similar literature? Is the inhibition effect in this study high or low compared to them?

9. The title is about antibacterial action, but antibacterial action is only the last part of the whole article, most of the context is about the characteristics of the extract. It is suggested that the title be modified to cover the entire article.

Author Response

Reviewer #1:

Comments:

In this manuscript, extract of Momordica charantia leaves and its microemulsions were obtained and the characteristics were studied including FTIR, spectroscopy, particle size, zeta potential, thermal stress, pH, electrical Conductivity, TEM, and antibacterial analysis. The experimental design and result analysis are logical. The overall work is robust and interesting. However, here are some suggestions below to improve the manuscript:

  1. Line 35-40, what is the connection between the first paragraph of the introduction and the title article and the following part? No logical relationship seems to be expressed.

ANSWER: We are very grateful to the honorable reviewer for their highly useful comments and suggestions.  The introduction began by emphasizing the organism to be fought in work with the use of  Momordica charantia, and then, in the third paragraph, Lines 47-50, make the connection with the title of the work.

  1. Line 70-80, paragraphs 6-7 of the introduction can be merged into one paragraph, and a few more examples of Momordica charantia leaves extract used in the microemulsion system can be given to show the significance of this study.

ANSWER: Solved

  1. Line 61-63, the authors state that Momordica charantia leaves extract has an antibacterial effect on some bacteria, but why only Staphylococcus aureus was selected in this manuscript? Why didn't it do an antibacterial effect on other bacteria?

ANSWER:  The reason was the division of labor; the research group was composed of three researchers, and each one was responsible for working with a different bacteria; the two choices were Escherichia coli and Enterobacter sp., which will soon be published in two more articles.

  1. Line 196 and Line 318, please revise the title of section 3.1 and 3.5, currently this statement is not suitable for this part of the topic.

ANSWER:  Solved

  1. Line 197-201, what results are indicated by the comparison of each curve in Figure 1?

ANSWER: The discussion of this result appears between lines 209-237. 

  1. Line 203 and 209, What does "graph 2" mean?

ANSWER: In fact, he misses referring to figure 1. This problem has already been resolved in the document.

  1. Line 244, why choose this key point and is there any basis for the choice of the location of this point?

ANSWER: For this study, it was necessary to obtain a two-phase Oil/Water system as proposed by the reference [28], Burch, G.E.; Winsor, T. THE PHLEBOMANOMETER: A NEW APPARATUS FOR DIRECT MEASUREMENT OF VANOUS PRESSURE IN LARGE AND SMALL VEINS. J. Am. Med. Assoc. 1943, 123, 91–92, doi:10.1001/JAMA.1943.82840370004006B. And from the ternary phase diagram (figure 2), this would only occur at this point, with 20% surfactant, 1% oil phase, and 79% water phase.

  1. Line 338-339, can the inhibition results in this section be compared to those in other studies, such as those cited here 7,12,36 or other similar literature? Is the inhibition effect in this study high or low compared to them?

ANSWER: Yes, these results can be compared to those of other studies, but to say whether it was higher or lower, we would have to submit our samples to the same reference tests7,12,36. What could not be done.

  1. The title is about antibacterial action, but antibacterial action is only the last part of the whole article, most of the context is about the characteristics of the extract. It is suggested that the title be modified to cover the entire article.

ANSWER: In response to your valuable suggestion, the title has been changed to: “The microemulsion with solubilization of the ethanolic extract of the leaves of Melão-de-São-Caetano (Momordica charantia)and antibacterial action.”

Reviewer 2 Report

In general, the manuscript is very interesting, only a few small details that I mark throughout the manuscript need to be attended to.

Author Response

Reviewer #2:

Comments:

  1. Have these leaves been identified or classified by a botanist? If so, please provide information.

ANSWER:  Unfortunately, we do not have this information, but the following link contains an official website with other important information about the Brazilian variety that was used in this work, information that we did not find necessary to include in it.

https://ferramentas.sibbr.gov.br/ficha/bin/view/especie/momordica_charantia

  1. Full details of the equipment, including country of origin.

ANSWER: Sorry, we don't have that information.

  1. What does this mean?

ANSWER: Solved.

  1. Translate into English.

ANSWER: Solved.

  1. Miniscule

ANSWER: Solved.

  1. Miniscule

ANSWER: Solved.

  1. No accent

ANSWER: Solved.

  1. Provide the ATTCC strain number or Code.

ANSWER: Please see line 190 of the document, this information appears right after the name of the bacteria and is in parentheses (AATCC25923).

Reviewer 3 Report

Dear authors, 

thank you for inviting me to participate in the examination of this work entitled:Antibacterial action of the microemulsion with solubilization of the ethanolic extract of the leaves of Melão-de-São-Caetano (Momordica charantia)

Title: the species name must be in italics

abstract: it is preferable to start with the objective of the work. the bacterial strains should be in italics

Introduction: the objective should be mentioned

material and methods

-The information on the identification of the plant and the name of the botanist who identifies this species should be added, this is very important.

-Bacteria should be italicized.

-Statistical analysis ???????????

- results and discussion 

This part is well presented, but the antibacterial capacity should be discussed as it is the most important part of this work.

-conclusion 

the results should not be discussed or mentioned in the conclusion, so that the summary is not copied. it is preferable to write perstectives.

Author Response

Reviewer #3:

Comments:

Dear authors, 

thank you for inviting me to participate in the examination of this work entitled: Antibacterial action of the microemulsion with solubilization of the ethanolic extract of the leaves of Melão-de-São-Caetano (Momordica charantia)

  1. Title: the species name must be in italics

ANSWER: Solved.

  1. Abstract: it is preferable to start with the objective of the work. the bacterial strains should be in italics.

ANSWER: Partially resolved; the abstract was not modified because articles published in this journal on which I based myself as a model present such a topic as it is, for example, A Molecular Docking Study Reveals That Short Peptides Induce Conformational Changes in the Structure of Human Tubulin Isotypes αβI, αβII , αβIII, and αβIV.

  1. Introduction: the objective should be mentioned.

ANSWER: See lines 81-85. I believe they understand this request.

Material and methods.

  1. -The information on the identification of the plant and the name of the botanist who identifies this species should be added, this is very important.

ANSWER: The name "scientific Momordica charantia" appears in the title and several other parts of the work; I believe that it meets this request. In Brazil, the country where the research was carried out, the most popular name is "Melon- de-São-Caetano."

5.-Bacteria should be italicized.

ANSWER: Solved.

  1. - Statistical analysis ???????????.

ANSWER: The ANOVA statistical analysis was considered for this work, but it was not carried out because a much larger number of samples would be needed in a short time for the defense of the master's thesis; this analysis will be done in another work that is complementary to this one. And which will be published soon.

  1. - Results and discussion 

This part is well presented, but the antibacterial capacity should be discussed as it is the most important part of this work.

ANSWER Solved:

Conclusion 

  1. The results should not be discussed or mentioned in the conclusion, so that the summary is not copied. it is preferable to write perstectives.

ANSWER: Solved.

Reviewer 4 Report

Dear Authors

After careful evaluation of the manuscript entitled "Antibacterial action of the microemulsion with solubilization of the ethanolic extract of the leaves of Melão-de-São-Caetano (Momordica charantia)", I regret to inform you that I do not find the manuscript suitable for publication in the Journal of Functional Biomaterials because it does not meet the novelty and impact requirements of the journal.

The work is not original and shows only incremental advance over prior research results. Publications in peer-reviewed journals are to disseminate knowledge. Therefore for a manuscript to be published in a well-recognized, international journal it must have significant scientific value.

The work is not original and does not contain new results that significantly advance the research field of functional biomaterials, antimicriobial activity of microemulsion and physico-chemical characterisation of the microemulsion.

Also, several similar research and review papers have been publisehd previsouly. Plese see below some of these articles :

1.       Brito, A. M. Q. D. (2020). Atividade contra staphylococcus aureus por sistema microemulsionado com extrato de momordica charantia solubilizado para potencial aplicação em têxtis esportivos (Master's thesis, Universidade Federal do Rio Grande do Norte).

2.       Trakoolthong, P., Ditthawuttikul, N., Sivamaruthi, B. S., Sirilun, S., Rungseevijitprapa, W., Peerajan, S., & Chaiyasut, C. (2022). Antioxidant and 5α-Reductase Inhibitory Activity of Momordica charantia Extract, and Development and Characterization of Microemulsion. Applied Sciences12(9), 4410.

3.       Patyal, P., & Thakur, N. (2019). Formulation and In-vitro evaluation of controlled polyherbal microemulsion for the treatment of diabetes mellitus. International Journal of Pharmacy & Life Sciences10(5).

4.       Stefan, C. S., Chitescu, C. L., Manolache, N. et al., (2022). The Investigation of Antimicrobial Activity of some Extracts from Momordica charantia by using as Solvent Extraction an Ionic Liquid. Farmacia70(1), 144-150.

5.       Ahmed, Z., & Noor, A. A. (2020). 22. Antibacterial activity of Momordica charantia L. and Citrus limon L. on gram positive and gram-negative bacteria. Pure and Applied Biology (PAB)9(1), 207-218.

6.       Ozusaglam, M. A., & Karakoca, K. (2013). Antimicrobial and antioxidant activities of Momordica charantia from Turkey. African Journal of Biotechnology12(13).

7.       Mada, S. B., Garba, A., Mohammed, H. A. A., Muhammad, A., Olagunju, A., & Muhammad, A. B. (2013). Antimicrobial activity and phytochemical screening of aqueous and ethanol extracts of Momordica charantia L. leaves. Journal of Medicinal Plants Research7(10), 579-586.

8.       Guarniz, W. A. S., Canuto, K. M., Ribeiro, P. R. V., Dodou, H. V., Magalhaes, K. N., Sá, K., & Bandeira, M. A. M. (2019). Momordica charantia L. Variety from northeastern Brazil: analysis of antimicrobial activity and phytochemical components. Pharmacognosy Journal11(6).

9.       Muribeca, A. D. J. B., Gomes, P. W. P., Paes, S. S., da Costa, A. P. A., Gomes, P. W. P., Viana, J. D. S., ... & da Silva, M. N. (2022). Antibacterial activity from Momordica charantia L. leaves and flavones enriched phase. Pharmaceutics14(9), 1796.

10.   Valizadeh, M., Beigomi, M., & Fazeli-Nasab, B. (2020). Antibacterial and Anti biofilm effects of ethanol and aceton leaf extract of Momordica charantia and Tecomella undulata against Acinetobacter baumanniiInt j adv biol biomed res8(4), 403-18.

FINAL DECISION = REJECT

Author Response

Reviewer #4:

Comments:

Dear Authors

After careful evaluation of the manuscript entitled "Antibacterial action of the microemulsion with solubilization of the ethanolic extract of the leaves of Melão-de-São-Caetano (Momordica charantia)", I regret to inform you that I do not find the manuscript suitable for publication in the Journal of Functional Biomaterials because it does not meet the novelty and impact requirements of the journal.

The work is not original and shows only incremental advance over prior research results. Publications in peer-reviewed journals are to disseminate knowledge. Therefore for a manuscript to be published in a well-recognized, international journal it must have significant scientific value.

The work is not original and does not contain new results that significantly advance the research field of functional biomaterials, antimicriobial activity of microemulsion and physico-chemical characterisation of the microemulsion.

Also, several similar research and review papers have been publisehd previsouly. Plese see below some of these articles :

  1. Brito, A. M. Q. D. (2020). Atividade contra staphylococcus aureus por sistema microemulsionado com extrato de momordica charantia solubilizado para potencial aplicação em têxtis esportivos(Master's thesis, Universidade Federal do Rio Grande do Norte).
  2. Trakoolthong, P., Ditthawuttikul, N., Sivamaruthi, B. S., Sirilun, S., Rungseevijitprapa, W., Peerajan, S., & Chaiyasut, C. (2022). Antioxidant and 5α-Reductase Inhibitory Activity of Momordica charantiaExtract, and Development and Characterization of Microemulsion. Applied Sciences12(9), 4410.
  3. Patyal, P., & Thakur, N. (2019). Formulation and In-vitroevaluation of controlled polyherbal microemulsion for the treatment of diabetes mellitus. International Journal of Pharmacy & Life Sciences10(5).
  4. Stefan, C. S., Chitescu, C. L., Manolache, N. et al., (2022). The Investigation of Antimicrobial Activity of some Extracts from Momordica charantia by using as Solvent Extraction an Ionic Liquid. Farmacia70(1), 144-150.
  5. Ahmed, Z., & Noor, A. A. (2020). 22. Antibacterial activity of Momordica charantia L. and Citrus limon L. on gram positive and gram-negative bacteria. Pure and Applied Biology (PAB)9(1), 207-218.
  6. Ozusaglam, M. A., & Karakoca, K. (2013). Antimicrobial and antioxidant activities of Momordica charantia from Turkey. African Journal of Biotechnology12(13).
  7. Mada, S. B., Garba, A., Mohammed, H. A. A., Muhammad, A., Olagunju, A., & Muhammad, A. B. (2013). Antimicrobial activity and phytochemical screening of aqueous and ethanol extracts of Momordica charantia L. leaves. Journal of Medicinal Plants Research7(10), 579-586.
  8. Guarniz, W. A. S., Canuto, K. M., Ribeiro, P. R. V., Dodou, H. V., Magalhaes, K. N., Sá, K., & Bandeira, M. A. M. (2019). Momordica charantia L. Variety from northeastern Brazil: analysis of antimicrobial activity and phytochemical components. Pharmacognosy Journal11(6).
  9. Muribeca, A. D. J. B., Gomes, P. W. P., Paes, S. S., da Costa, A. P. A., Gomes, P. W. P., Viana, J. D. S., ... & da Silva, M. N. (2022). Antibacterial activity from Momordica charantiaL. leaves and flavones enriched phase. Pharmaceutics14(9), 1796.
  10. Valizadeh, M., Beigomi, M., & Fazeli-Nasab, B. (2020). Antibacterial and Anti biofilm effects of ethanol and aceton leaf extract of Momordica charantiaand Tecomella undulata against Acinetobacter baumanniiInt j adv biol biomed res8(4), 403-18.

FINAL DECISION = REJECT

ANSWER:

Dear reviewer,

I would like to begin by thanking you for your valuable time in reviewing our work. However, we do not agree with your decision to reject it based on its lack of originality. All the results presented in our work are the outcome of meticulous research conducted by a diverse group of researchers, which culminated in an approved master's thesis, the first document on your list of ten works. Therefore, I respectfully question your decision.

As mentioned earlier, the first work on the list is the master's thesis from which the results for this article emerged. The other works, even though they may have used the same study material (Momordica charantia), had different applications or slightly similar results, and for this reason, we used them as references in some justifications or are too old, such as more than five years since their publication (Reference 6).

I would like to conclude by stating that science works in this way; there is always a starting point, and the same study material can be used for different purposes. For example, investigating whether Titanium, which is used in space on the International Space Station or rockets, would have applications in orthopedic implants, led to their existence. Similarly, investigating whether Momordica charantia, which was used to treat Diabetes Mellitus, would have other applications with different formulations.

Respectfully,

Dr. Ivan Alves De Souza.

 (Corresponding Author)

Round 2

Reviewer 1 Report

Most of the problems have been solved, except for the presence of "graph 2" in line 272.